# Traditional Chinese medicine Lianhua Qingwen treating corona virus disease 2019 (COVID-19): Meta-analysis of randomized controlled trials

Mengjie Zeng[1], Linjun Li[2], Zhiquan Wu[3]*

1 Department of Traditional Chinese Medicine, Macau University of Science and Technology, Macau, China, 2 Department of Acupuncture and Tuina, Yunnan University of Traditional Chinese Medicine, Kunming, China, 3 Department of Rehabilitation Medicine, Sanya Central Hospital, Sanya, China

* 513376829@qq.com

**Data Availability Statement:** All relevant data are within the manuscript and its Supporting Information files.

## Abstract

### Introduction

As the global epidemic continues to spread, countries have tapped effective drugs to treat new coronavirus pneumonia. The therapeutic effect of the traditional Chinese medicine Lianhua Qingwen in this new coronary pneumonia epidemic has attracted attention from all walks of life, and relevant research reports continue to appear. Therefore, we conducted a systematic review of the clinical efficacy and safety of the traditional Chinese medicine Lianhua Qingwen in the treatment of new coronavirus pneumonia (COVID-19) (referred to as "new coronary pneumonia"), and evaluated the overall level of research quality.

### Methods

We searched seven databases and retrieved the Chinese Journal Full-text Database (CNKI), Vip Database (VIP), China Biomedicine (SinoMed), Wanfang Database and PubMed, Cochrane Central, EMBASE from October 2019 to May 2020 Literature references. We included randomized controlled trials (RCTs) that tested the efficacy of the traditional Chinese medicine lotus clearing plague in the treatment of new coronavirus pneumonia. The authors extracted data and independently assessed quality. We used Stata15.1 software to analyze the data of randomized trials.

### Results

A total of 2 articles were identified, including 154 patients. All the participating patients were diagnosed with new coronavirus pneumonia (COVID-19). The meta-analysis results showed that the disappearance rate of the main clinical symptoms of Chinese medicine Lianhua Qingwen in the treatment of new coronavirus pneumonia was significantly higher than that of the control group [OR = 3.34, 95% CI (2.06, 5.44), P <0.001]; the disappearance rate of other clinical secondary symptoms is significantly higher than the control group [OR = 6.54, 95% CI (3.59, 11.90), P <0.001]. The duration of fever was significantly lower than that of the control group [OR = -1.04, 95% CI (-1.60, -0.49), P <0.001]. It is confirmed that

**Funding:** This work was supported by Sanya medical science and technology innovation project, No.: 2017YW06

**Competing interests:** The authors have declared that no competing interests exist.

**Abbreviations:** CNKI, China Network Knowledge Infrastructure; MD, Mean difference; RCT, Random Controlled Trial; RR, Risk ratio; SD, Standard deviation; VIP, Chinese Scientific Journal Database; OR, odds ratio.

the traditional Chinese medicine Lianhua Qingwen treatment improves the clinical effectiveness, and also has certain advantages in relieving cough and fever.

## Conclusion

The treatment of new pneumonia with traditional Chinese medicine lotus clearing plague can be used as an effective therapy to improve the clinical symptoms of new coronary pneumonia. More rigorous design, multi-center, and prospective RCTs are necessary to further determine the effectiveness and safety of the traditional Chinese medicine lotus decoction in the treatment of new pneumonia.

## Introduction

The new coronavirus pneumonia (COVID-19), which has the characteristics of extremely contagious, long incubation period, diverse clinical manifestations, and a wide range of susceptible people [1]. If treatment is not timely, it is easy to cause death. This disease belongs to the category of "warm disease" in traditional Chinese medicine, which is highly contagious and epidemic, and can also belong to "warm disease" [2]. In the fight against the COVID-19 epidemic, China has always adhered to the treatment principle of "emphasis on both Chinese and Western medicine", and the traditional Chinese medicine lotus clearing has played an important role in the fight against the epidemic [3]. Studies have shown that Lianhua Qingwen granules have a good effect on respiratory diseases induced by various influenza viruses, and its mechanism of action includes strengthening the body's immunity, inhibiting inflammation of the respiratory tract, and reducing inflammation damage in the lungs [4].

Up to now, there is no special drug for the treatment of new coronary pneumonia. At the same time, the cumulative number of diagnoses worldwide has reached 3 million, and the number of deaths has reached more than 200,000, which has already imposed a heavy burden on the global economy and people's lives. Countries have also carried out clinical trials and vaccine research and development, with a view to finding therapeutic drugs as soon as possible. In our country, Chinese medicine has very successful experience in preventing and treating epidemic diseases. Its Lianhua Qingwen is made from the ancient prescription Yin Qiao San and Ma Xing Shi Gan Decoction. Its main components are: Forsythia, Honeysuckle, Ephedra (Branch), Bitter Almond (Fried), Gypsum, Banlangen, Mianma It consists of 13 herbs such as Guanzhong, Houttuynia cordata, patchouli, rhubarb, Rhodiola rosea, menthol, and licorice [5]. Common formulations of this formula are capsules, granules, and decoctions. Through the bibliometrics study of modern literature and the analysis of the names of the diseases treated by Lianhua Qingwen, it is found that the traditional Chinese medicine diseases with pulmonary fever are the most frequent.

In order to scientifically evaluate the efficacy and effectiveness of the traditional Chinese medicine lotus clearing plague in the treatment of new coronavirus pneumonia, we comprehensively reviewed the medical literature and conducted a meta-analysis of the randomized controlled trials of the traditional Chinese medicine Lianhua clearing plague in the treatment of new coronavirus pneumonia.

## Methods

### Search strategy

Three English electronic databases and four Chinese electronic databases were searched: PubMed, Cochrane Central, EMBASE, China National Knowledge Infrastructure (CNKI),

China Biomedicine (SinoMed), China Science Journal Database (VIP) and Wanfang Database. Conference records and papers were also retrieved from CNKI and Wanfang databases for unpublished trials. Depending on the search database, the following search terms (or Chinese database equivalent to Chinese) are used: "Lianhua Qingwen Capsule", "Lianhua Qingwen Capsule", "Lianhua Qingwen Granule", "New Coronavirus Pneumonia", "COVID-19", "NCP", "New Coronary Pneumonia", "Coronavirus", "corona virus disease 2019" and "Random".

## The specific search strategy of PubMed is as follows

# 1 Search (((((((((((Coronavirus disease [Title / Abstract]) OR COVID-19 [Title / Abstract]) OR Corona virus disease 2019

 # 2 Search (((((Lianhua Qingwen [Title / Abstract]) OR Lianhua Qingwen Capsule [Title / Abstract]) OR Lianhuaqingwen particles [Title / Abstract])

 # 3 Search random

 # 1and # 2 and # and3

## Eligibility criteria

**Inclusion and exclusion criteria.** For the new coronary pneumonia standard, refer to the "New Coronavirus Pneumonia Diagnosis and Treatment Program" (trial seventh edition) [5,6]. (1) Type of study: randomized controlled trials or non-randomized studies. (2) Research objects: Chinese patients with new coronavirus pneumonia. (3) Intervention measures: the treatment group was treated with Lianhua Qingwen combined with conventional therapy, and the control group was treated with conventional therapy.

Exclusion criteria: (1) Patients who are not clearly identified as new coronaviruses; (2) Student cannot represent samples of the general population; (3) There are repeated reports and incomplete information.

## Research options

An author (LJL) independently screened the literature. First of all, the title and abstract of all records were screened for relevance, and then it was determined whether the full text of the relevant research may be eligible. If there is a disagreement, it is resolved through discussion with the author (MJZ). The extracted data include: first author, study time, study location, diagnostic criteria, inclusion criteria, exclusion criteria, number and design of cases in the treatment group and control group, and record the disappearance rate and duration of main symptoms (fever, cough) in the treatment group and control group Time and efficiency (Table 1).

## Assessment of risk of bias

An author (LJLand ZQW) used the Cochrane bias risk tool to independently assess the quality of included trials [7]. The following items were evaluated: the following 7 aspects were evaluated: (1) random sequence generation; (2) allocation concealment; (3) patient blindness; (4) evaluator

**Table 1. Summary of studies involved in the meta-analysis.**

| Author | Published Time | Sample Size (T/C) | Intervention measure | | Outcome |
|---|---|---|---|---|---|
| | | | T | C | |
| Kaizhong Yao [8] | 2020 | 21/21 | Lian hua Qing wen and Conventional treatment | Conventional treatment | ①+②+③ |
| Dezhong Cheng [9] | 2020 | 51/51 | Lian hua Qing wen and Conventional treatment | Conventional treatment | ①+②+③ |

T: Treatment Group, C: Control Group; ①: Disappearance rate of cardinal symptoms; ②: Duration of fever; ③: Disappearance rate of other symptoms.

blindness; (5) incomplete data (6) Selective reporting; (7) Other sources of bias. According to the above criteria, the evaluation results are ranked with low risk, unclear risk or high risk. When we judge other biases, we will consider sample size estimates, inclusion and exclusion criteria. The differences were resolved through discussion with the author (MJZ) (Table 2).

## Data analysis

We use the software Stata15.1 for Meta analysis. Heterogeneity evaluation was conducted on the results of each study, and the χ2 test was adopted uniformly. If $I^2 \geq 50\%$, it means that the existence of each study group means that the heterogeneity between the study groups is low, and the fixed effect mode is used. In all the above analysis, the measurement data adopts standardized mean difference (SMD) or mean difference (MD) for effect combination, and the count data uses relative risk (RR) or odds ratio (OR) to combine effect amounts. Inverted funnel charts were used to conduct publication bias risk assessments for each study.

## Results

### Search results

According to the retrieval strategy, we retrieved 27 articles from 1 electronic databases. A total of 26 duplicate clauses were excluded. After preliminary screening of titles and abstracts, 2 articles were excluded because they did not meet the research criteria. We searched 24 full-text studies to further identify the reasons why 20 articles were deleted. Finally, 2 participated in the Meta analysis (Fig 1).

### Study characteristics

(Table 1) summarizes the characteristics of 2 trials involving 142 participants. All included trials were conducted in China and published in Chinese. The sample size varied from 20 to 50 participants. All participants were diagnosed by clinical manifestations and CT or MRI examinations, and the efficacy was evaluated at the end of treatment.

Among the 2 qualified RCTs, the clinical primary symptom disappearance rate measurement, secondary symptom disappearance rate evaluation, and concurrent fever duration were evaluated. Therefore, it reflects the therapeutic effect of the traditional Chinese medicine Lianhua Qingwen on the treatment of (COVID-19).

### Continuous data outcomes

We will involve 142 patients for a comprehensive analysis. According to the disappearance rate of cardinal symptoms (fever, cough, fatigue), Meta analysis results are shown in (Fig 2).

**Table 2. Quality of the included studies assessed by Cochrane risk of bias assessment.**

| Quality assessment criteria | Kaizhong Yao, 2020 | Dezhong Cheng, 2020 |
|---|---|---|
| Random sequence generation(selection bias) | + | + |
| Allocation concealment(selection bias) | + | + |
| Blinding participants and personnel(performance bias) | - | - |
| Blinding of outcome assessment(detection bias) | + | + |
| Incomplete outcome data(attrition bias) | + | + |
| Selective reporting(reporting bias) | + | + |
| Other bias | + | + |
| Overall quality score (maximum = 7) | 6 | 6 |

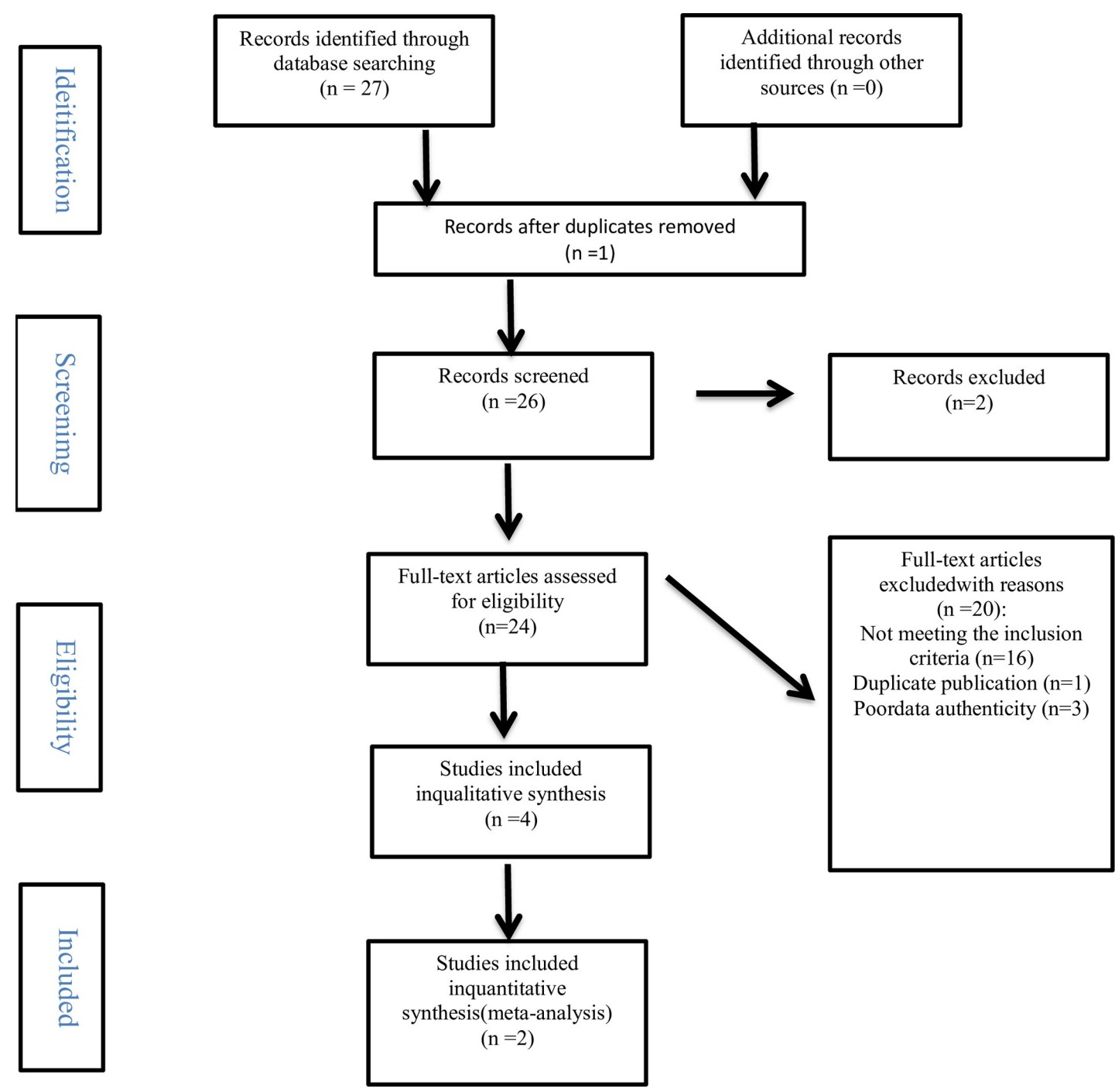

**Fig 1. Study selection flow chart.**

Chi$^2$ test shows that I$^2$ = 0% (<50%), P = 0.745. It shows that there is no heterogeneity between the included documents, so we choose to use a fixed effect model for combined analysis. It can be seen from the forest map that the disappearance rate of the main clinical symptoms of the new coronavirus pneumonia treated with the traditional Chinese medicine Lianhua Qingwen is significantly higher than that of the control group [OR = 3.34, 95% CI (2.06, 5.44), P <0.001].

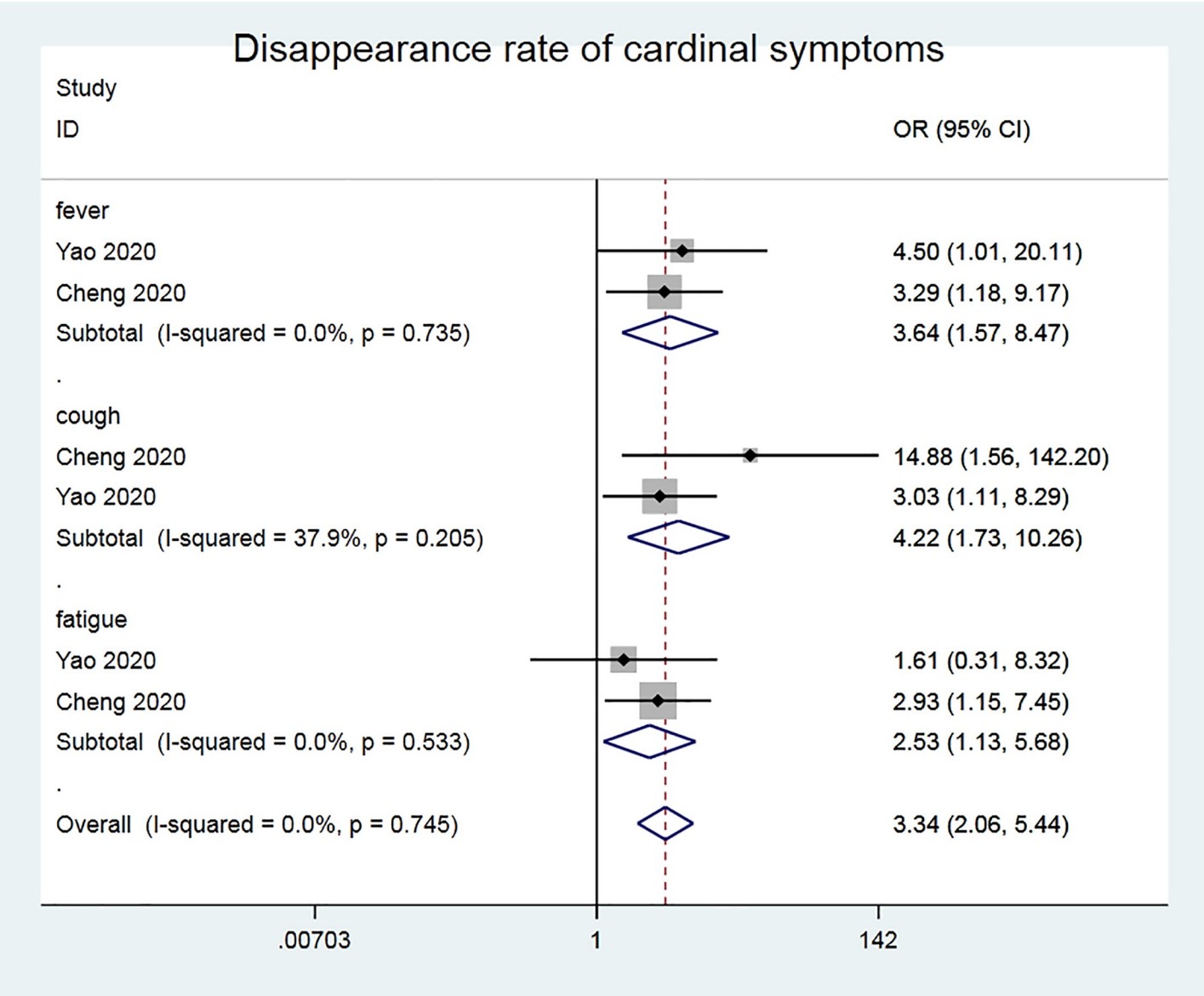

**Fig 2. Disappearance rate of cardinal symptoms.**

According to the disappearance rate of other symptoms (muscle pain, sputum, nasal congestion, runny nose, chest tightness, difficulty breathing, nausea and vomiting, loss of appetite), Meta analysis results are shown in (Fig 3), Chi$^2$ test shows that I$^2$ = 0% (<50%), P = 0.574 choose to use fixed effect model for combined analysis. It can be seen from the forest map that the disappearance rate of other secondary symptoms in the clinical application of the traditional Chinese medicine Lianhua Qingwen in the treatment of new coronavirus pneumonia is significantly higher than that in the control group [OR = 6.54, 95% CI (3.59, 11.90), P <0.001].

According to the disappearance rate of fever duration, Meta analysis results are shown in (Fig 4). Chi$^2$ test shows that I$^2$ = 0% (<50%), P = 0.623, indicating that there is no heterogeneity between the included literatures, so the random effect model was chosen analysis. It can be seen from the forest map that the duration of lung fever with the Chinese medicine Lianhua

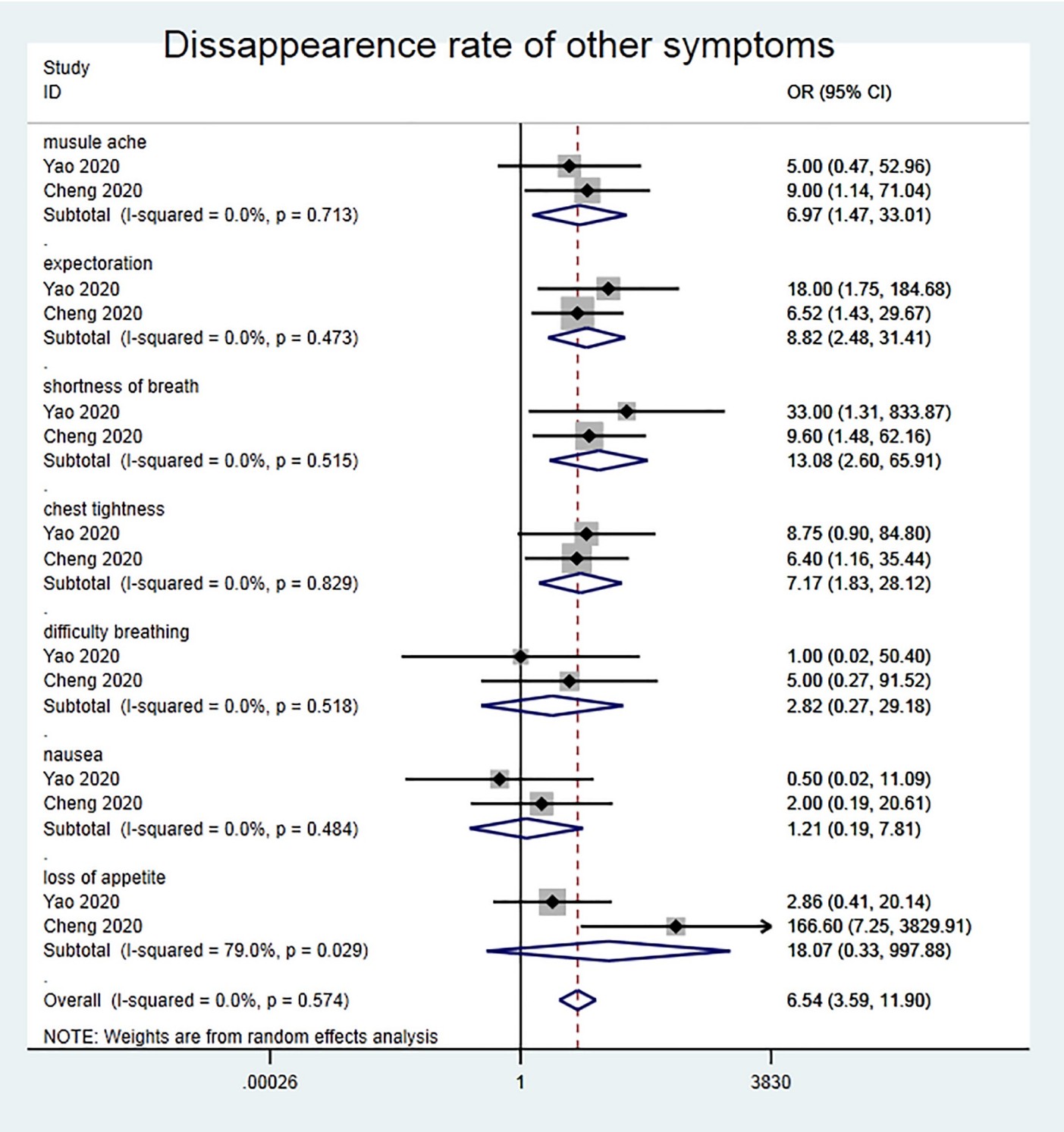

**Fig 3. Disappearance rate of other symptoms.**

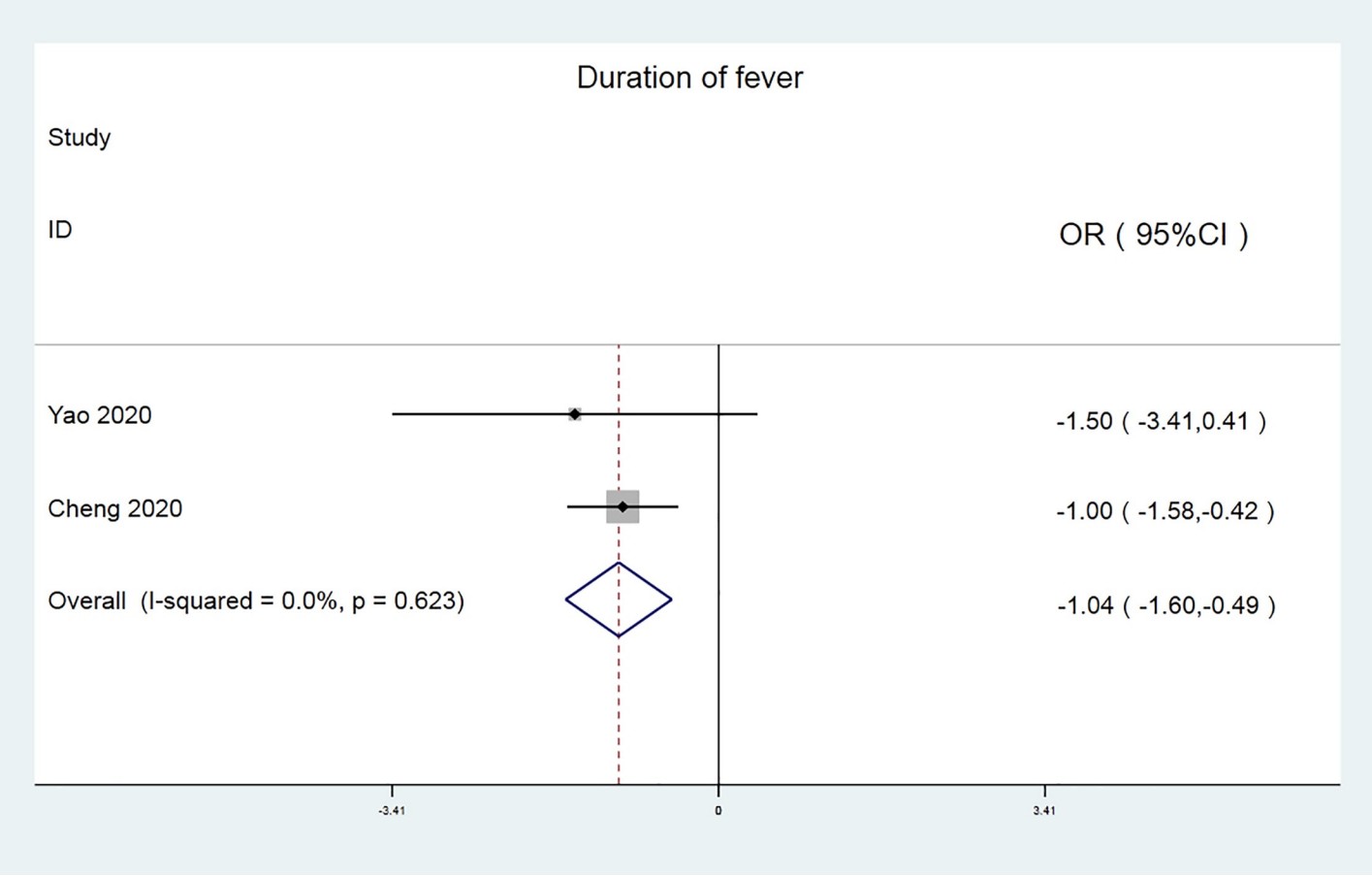

**Fig 4. Disappearance rate of fever duration.**

Qingwen in the treatment of new coronavirus is significantly lower than that in the control group [OR = -1.04, 95% CI (-1.60, -0.49), P <0.001].

## Funnel plot

The funnel chart was used to analyze the two reported clinical main symptoms trials to explore the bias (Fig 5). The plot is symmetrical, indicating no obvious deviation.

Due to the large number of descriptions of clinical secondary symptoms, Egger was used to analyze two reported trials of clinical secondary symptoms to explore their bias (Fig 6), all appear near the moving average, indicating that there is no obvious deviation.

The funnel chart was used to analyze the two reported fever time duration experiments to explore the bias (Fig 7). The plot is symmetrical, indicating no obvious deviation.

## Discussion

### Main findings

Among the existing clinical studies, there are relatively few clinical studies on the treatment of new-onset coronary heart disease with traditional Chinese medicine intervention, and the number of studies is considerable. The clinical evidence obtained during the fight against the SARS epidemic in 2003 and the fight against the influenza A epidemic in 2009 also shows that

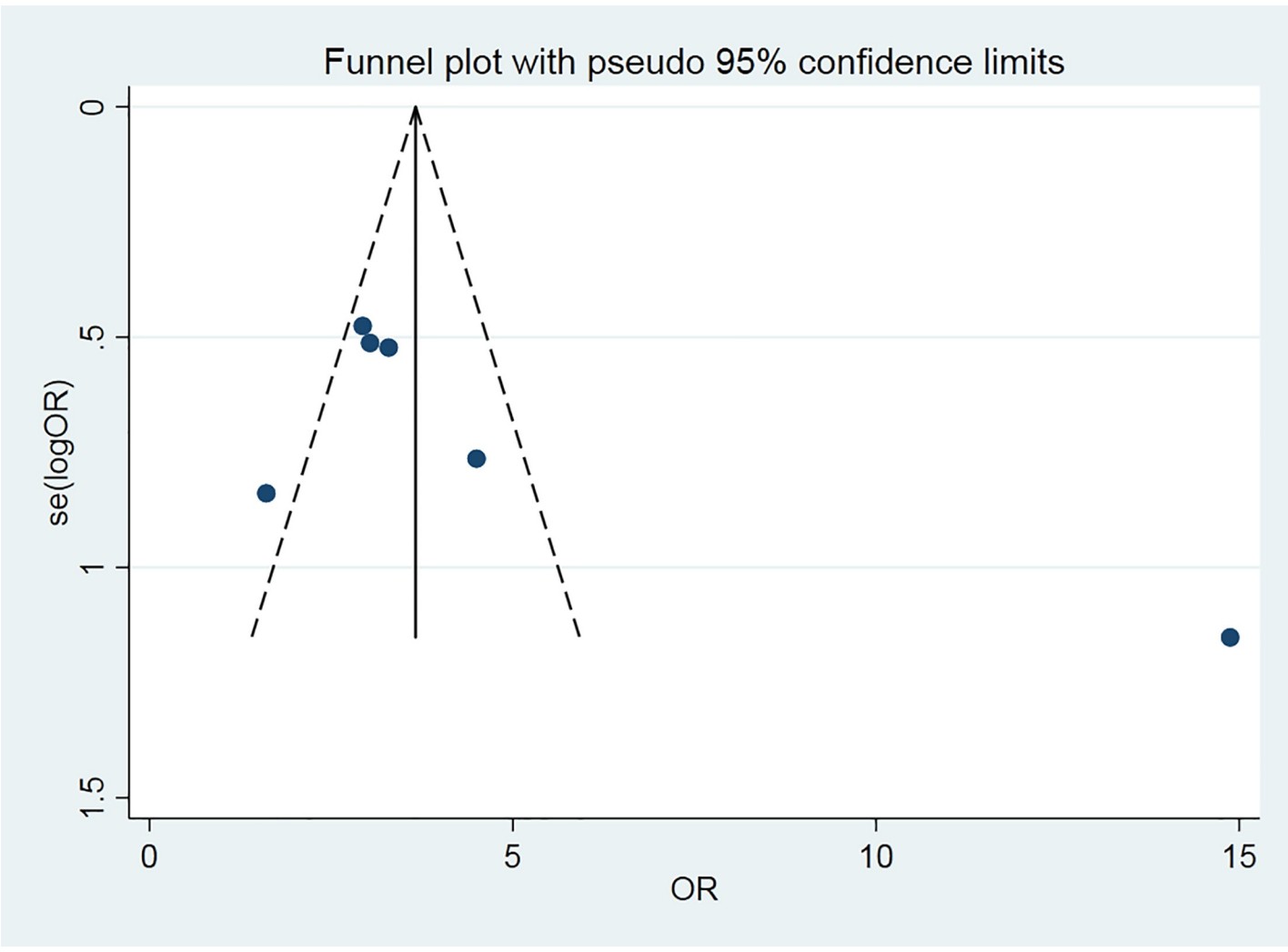

**Fig 5. Clinical main symptoms trials.**

the traditional Chinese medicine Lotus qingwen has played an important role in the prevention and control of viral public health events and has certain clinical application value [8]. One of the studies showed that the improvement rate of CT can also explain its therapeutic effect [9]. However, at present, there are few studies that can be used for comprehensive analysis to further exert its research value, and the quality of evidence is not high. Even take antipyretic capsule has anti-inflammatory, antiviral, regulate immune function [10], is widely used in the prevention and treatment of viral flu [11], some scholars study [12, 13], even take antipyretic in the treatment of COVID—19, through multiple targets, multiple components, in the role of multiple pathways coronavirus, even take antipyretic capsule main ingredients with Mpro, ACE2 has good combining ability and action mechanism may be related to antivirus, immune regulation, etc. Although there are few literatures in this study, the analysis results show that novel Coronavirus pneumonia with lotus Clearing away fever has a definite therapeutic effect, and the therapeutic effect of Lianhua clearing away fever capsule is significantly better than that of the control group in improving the duration of fever, disappearance rate of major symptoms and disappearance rate of minor symptoms.

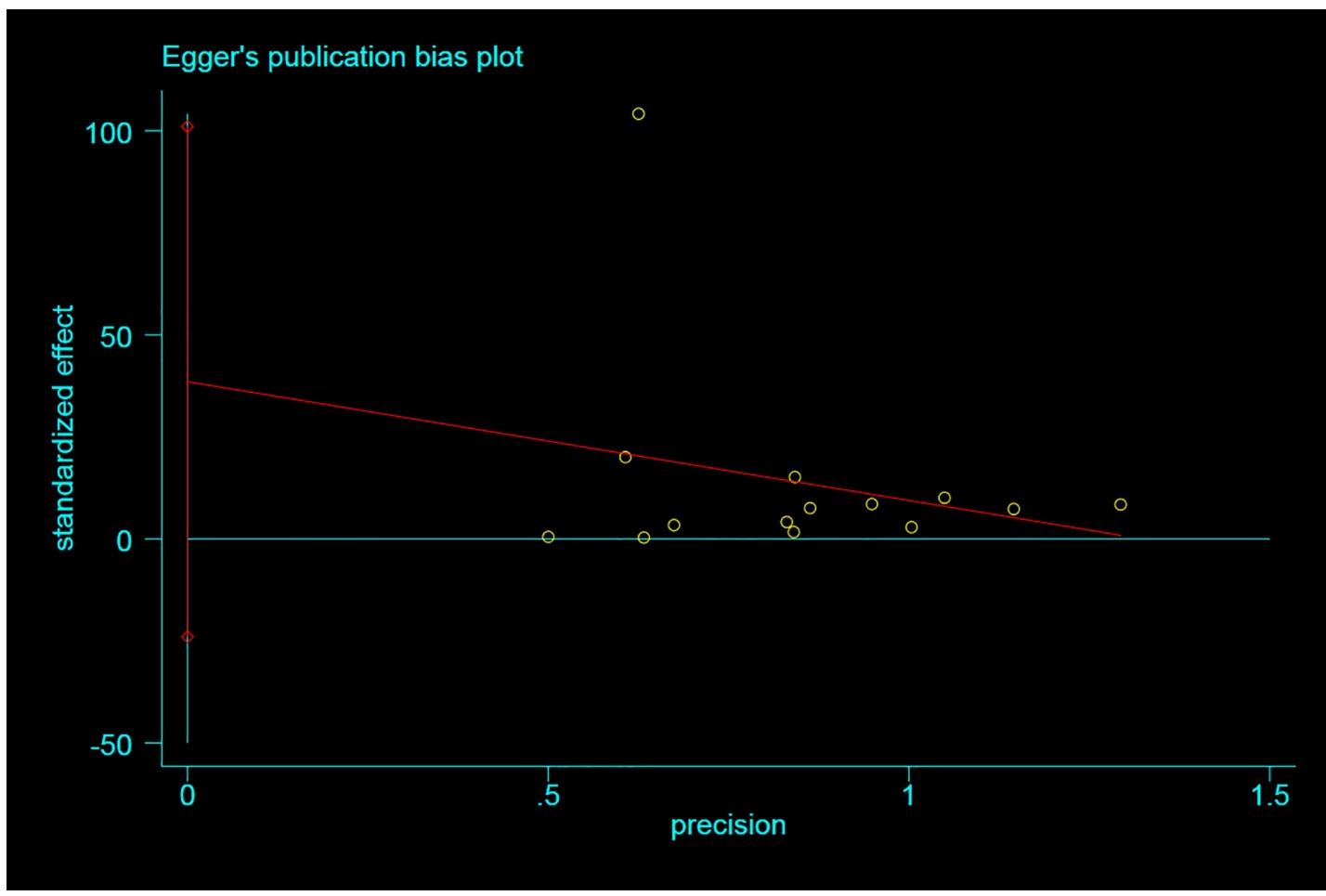

**Fig 6. Descriptions of clinical secondary symptoms.**

## Strengths and limitations

Our research also has some limitations. The methodological quality of the included trials is usually poor. Most trials did not report randomization procedures, and all trials lacked blinding information. In addition, no intention analysis and pre-review sample size estimation were conducted. Due to the insufficient number of trials included in the meta-analysis, there may be potential publication bias [14]. Due to the large amount of unexplained heterogeneity, the performance of some meta-analyses is limited.

## Implications for further study

In further research, we emphasized four issues that should be considered in TCM research: 1) The sample size should be based on sufficient statistics and the calculation of the sample size method should be reported in the text; the randomization method needs to be fully described, And appropriate hiding; baseline information should be reported in detail; 2) some drugs with unclear control effects: some conventional drugs are ineffective or harmful; 3) future trials should pay more attention to adverse events, especially the long-term safety of treatment investigations Adverse events should be recorded and reported using international standard medical terminology. 4) The mechanism and mechanism of traditional Chinese medicine treatment of new coronary pneumonia should be increased.

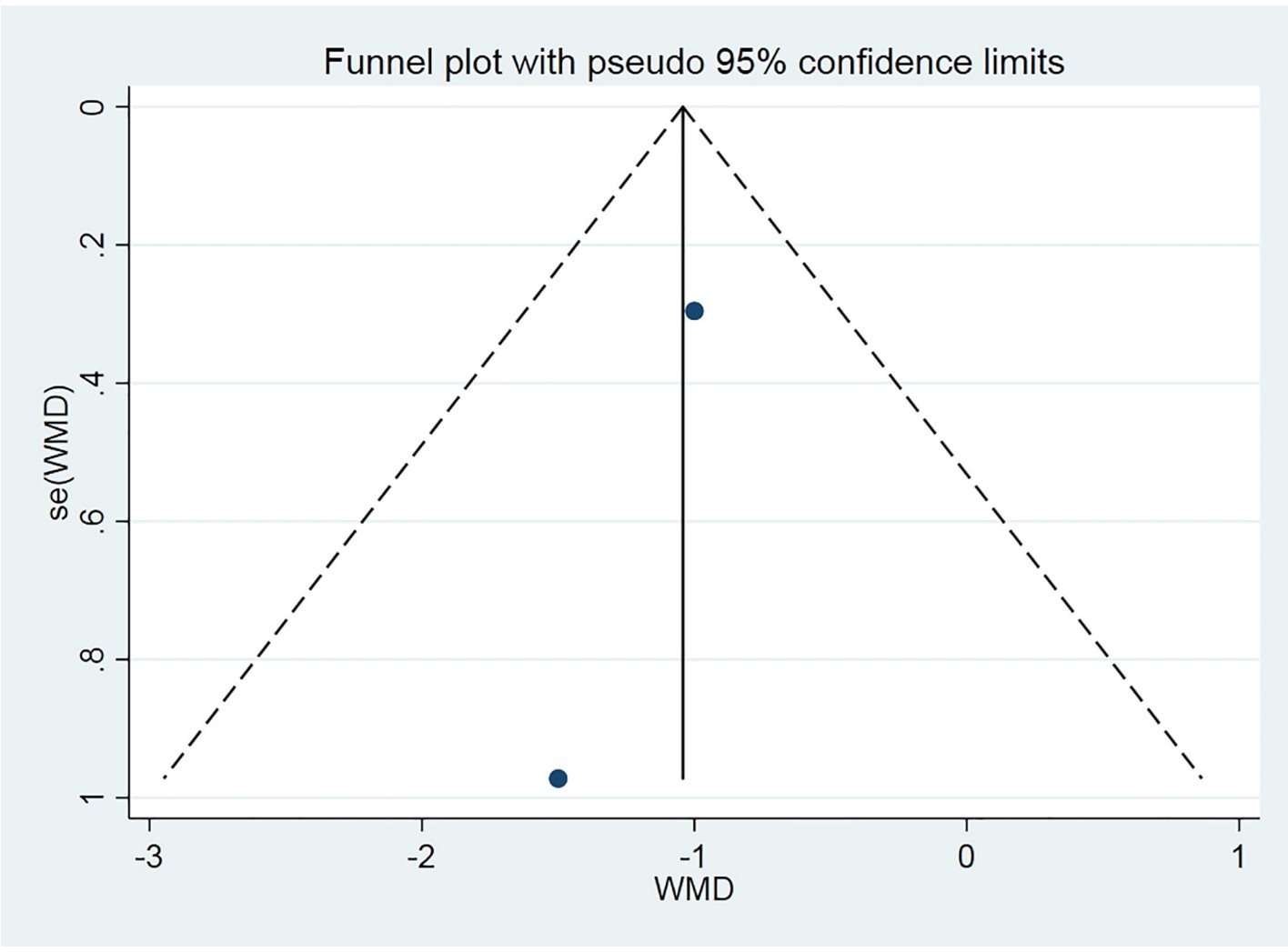

**Fig 7. Fever time duration.**

## Conclusions

The results of this test suggest that it is effective, and it can be confirmed that the traditional Chinese medicine Lianhua Qingwen is effective in treating new coronavirus pneumonia. However, due to the poor quality of the method and the small amount of literature, it is not possible to determine how effective it is. We recommend conducting more well-designed large-sample clinical trials, which can reduce the one-sidedness, chance, and heterogeneity between studies in a single article. At the same time, the safety, effectiveness, and Reliability makes a more scientific, objective and reasonable evaluation.

## Supporting information

**S1 Checklist. PRISMA 2009 checklist.**
(DOC)

**S1 File.**
(DOC)

## Author Contributions

**Formal analysis:** Linjun Li.

**Funding acquisition:** Zhiquan Wu.

**Investigation:** Linjun Li.

**Writing – original draft:** Mengjie Zeng, Linjun Li.

**Writing – review & editing:** Mengjie Zeng, Zhiquan Wu.

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
