## [Decision Letter · Decision Letter 0]

16 Jul 2020

PONE-D-20-12744

Traditional Chinese Medicine Lianhua Qingwen Treating corona virus disease 2019(COVID-19):meta-analysis of randomized controlled trials

PLOS ONE

Dear Dr. Wu,

Thank you for submitting your manuscript to PLOS ONE. After careful consideration, we feel that it has merit but does not fully meet PLOS ONE’s publication criteria as it currently stands. Therefore, we invite you to submit a revised version of the manuscript that addresses the points raised during the review process.

The manuscript will be reconsidered after revision. Please amend your manuscript according to reviewers' suggestions.

We look forward to receiving your revised manuscript.

Kind regards,

Prof. Raffaele Serra, M.D., Ph.D

Academic Editor

PLOS ONE

Journal Requirements:

5.We suggest you thoroughly copyedit your manuscript for language usage, spelling, and grammar. If you do not know anyone who can help you do this, you may wish to consider employing a professional scientific editing service.  

Additional Editor Comments (if provided):

The manuscript is potentially interesting. Please amend the manuscript according to reviewers' suggestions and resubmit.

Reviewers' comments:

Reviewer's Responses to Questions

**Comments to the Author**

1. Is the manuscript technically sound, and do the data support the conclusions?

Reviewer #1: Partly

2. Has the statistical analysis been performed appropriately and rigorously? 

Reviewer #1: Yes

3. Have the authors made all data underlying the findings in their manuscript fully available?

Reviewer #1: Yes

4. Is the manuscript presented in an intelligible fashion and written in standard English?

Reviewer #1: Yes

5. Review Comments to the Author

Reviewer #1: Tha authors conducted a systematic review of the clinical efficacy and safety of the

traditional Chinese medicine Lianhua Qingwen in the treatment of new coronavirus

pneumonia (COVID-19), and evaluated the overall level of research quality

The manuscript is novel but the main problem is that it lacks of critical insight and the discussion is very short. I would suggest to expand this section a little bit deepening your critical insight.

6. PLOS authors have the option to publish the peer review history of their article (what does this mean?). If published, this will include your full peer review and any attached files.

Reviewer #1: No

---

## [Author Response · Author response to Decision Letter 0]

16 Aug 2020

Reviewer #1:

1.Is the manuscript technically sound, and do the data support the conclusions?

The application of meta-analysis avoids the limitations of a single small-sample clinical trial, makes the analysis results more comprehensive and reliable, and provides a good basis for medical decision-making. All data of Lianhua Qingwen's treatment of new coronary pneumonia are based on strict screening criteria to summarize and analyze published papers. However, the data of the selected two papers are based on clinical treatment. So the conclusion is reliable and reasonable.Thanks.

Reviewer #2:

2.Has the statistical analysis been performed appropriately and rigorously?

All data processing is carried out in accordance with strict standards, and it is also the conclusion that the three authors jointly discussed and decided to use special statistical tools.Thanks.

Reviewer #3:

3.Have the authors made all data underlying the findings in their manuscript fully available?

At that time, all the data were strictly screened and provided.Thanks.

Reviewer #4:

4.Is the manuscript presented in an intelligible fashion and written in standard English?

Without any plagiarism, we will be responsible for the originality of the manuscript. At the same time, I have consulted the editor of the magazine before. If necessary, you can forward the content of the email. At the same time, if there is plagiarism, our author will take the initiative to withdraw the manuscript immediately, and the reviewers are also asked to respect our results. All languages are carefully written and checked many times. If there are some problems, please point them out. Thanks.

Reviewer #5:

We have added some reasonable comments in the discussion section, and at the same time added a description that fits the article based on the actual situation of the manuscript.

---

## [Decision Letter · Decision Letter 1]

26 Aug 2020

Traditional Chinese Medicine Lianhua Qingwen Treating corona virus disease 2019(COVID-19):meta-analysis of randomized controlled trials

PONE-D-20-12744R1

Dear Dr. Wu,

We’re pleased to inform you that your manuscript has been judged scientifically suitable for publication and will be formally accepted for publication once it meets all outstanding technical requirements.

Kind regards,

Prof. Raffaele Serra, M.D., Ph.D

Academic Editor

PLOS ONE

Additional Editor Comments (optional):

amended manuscript is acceptable

Reviewers' comments:

Reviewer's Responses to Questions

**Comments to the Author**

1. If the authors have adequately addressed your comments raised in a previous round of review and you feel that this manuscript is now acceptable for publication, you may indicate that here to bypass the “Comments to the Author” section, enter your conflict of interest statement in the “Confidential to Editor” section, and submit your "Accept" recommendation.

Reviewer #1: All comments have been addressed

2. Is the manuscript technically sound, and do the data support the conclusions?

Reviewer #1: Yes

3. Has the statistical analysis been performed appropriately and rigorously? 

Reviewer #1: Yes

4. Have the authors made all data underlying the findings in their manuscript fully available?

Reviewer #1: Yes

5. Is the manuscript presented in an intelligible fashion and written in standard English?

Reviewer #1: Yes

6. Review Comments to the Author

Reviewer #1: The manuscript has been properly amended. Now it is ready for publication. No further changes are needed.

7. PLOS authors have the option to publish the peer review history of their article (what does this mean?). If published, this will include your full peer review and any attached files.

Reviewer #1: No

---

## [Editor Report · Acceptance letter]

2 Sep 2020

PONE-D-20-12744R1 

Traditional Chinese Medicine Lianhua Qingwen Treating corona virus disease 2019(COVID-19):meta-analysis of randomized controlled trials 

Dear Dr. Wu:

I'm pleased to inform you that your manuscript has been deemed suitable for publication in PLOS ONE. Congratulations! Your manuscript is now with our production department. 

Kind regards, 

on behalf of

Prof. Raffaele Serra 

Academic Editor

PLOS ONE